# An ancestral SARS-CoV-2 vaccine induces anti-Omicron variants antibodies by hypermutation

Seoryeong Park [1,2,10], Jaewon Choi [3,4,10], Yonghee Lee[5], Jinsung Noh[5,6], Namphil Kim[5], JinAh Lee [7], Geummi Cho [1,8], Sujeong Kim[1,8], Duck Kyun Yoo [1,8], Chang Kyung Kang[9], Pyoeng Gyun Choe[9], Nam Joong Kim[9], Wan Beom Park [9], Seungtaek Kim [7] ✉, Myoung-don Oh [9] ✉, Sunghoon Kwon [3,5,6] ✉ & Junho Chung [1,2,8] ✉

The immune escape of Omicron variants significantly subsides by the third dose of an mRNA vaccine. However, it is unclear how Omicron variant-neutralizing antibodies develop under repeated vaccination. We analyze blood samples from 41 BNT162b2 vaccinees following the course of three injections and analyze their B-cell receptor (BCR) repertoires at six time points in total. The concomitant reactivity to both ancestral and Omicron receptor-binding domain (RBD) is achieved by a limited number of BCR clonotypes depending on the accumulation of somatic hypermutation (SHM) after the third dose. Our findings suggest that SHM accumulation in the BCR space to broaden its specificity for unseen antigens is a counterprotective mechanism against virus variant immune escape.

Since the emergence of severe acute respiratory syndrome coronavirus 2 (SARS-CoV-2), over 14 million sequences of variants have been collected and shared via the Global Initiative on Sharing All Influenza Data (GISAID)[1]. While most mutations have little effect or are detrimental to the virus, a small subset of mutations may provide a selective advantage leading to a higher reproductive rate[2]. The spike protein, a viral coat protein that mediates viral attachment to host cells and fusion between the virus and the cell membrane, is the primary target of neutralizing antibodies[3]. Serological analysis has shown that the receptor-binding domain (RBD) of the spike protein is the target of 90% of neutralizing activity in the immune sera[4,5]. In this context, the RBD has become the essential component of most mRNA-, adenovirus-, and recombinant protein-based vaccines[6].

However, Omicron variant BA.1 has accumulated 15 mutations in RBD[7], resulting in a 22-fold reduction in neutralization by plasma from vaccinees receiving two doses of the BNT162b2 vaccine[7]. Although bivalent vaccines have been developed to overcome the immune evasion of Omicron variant[8–11], the majority of the population has received only monovalent vaccines to date. Fortunately, it has been proven that a third dose of the BNT162b2 monovalent vaccine can neutralize BA.1 and several recent studies have demonstrated a general increase in somatic hypermutation(SHM) of virus-specific antibodies after the third dose[12–18]. However, the mechanism by which Omicron variant-neutralizing antibodies are generated through repeated exposure to the ancestral spike protein remains unclear. In this study, we analyzed the chronological B-cell receptor

[1]Department of Biochemistry and Molecular Biology, Seoul National University College of Medicine, Seoul, Republic of Korea. [2]Interdisciplinary Program in Cancer Biology Major, Seoul National University College of Medicine, Seoul, Republic of Korea. [3]Interdisciplinary Program in Bioengineering, Seoul National University, Seoul, Republic of Korea. [4]Integrated Major in Innovative Medical Science, Seoul National University, Seoul, Republic of Korea. [5]Department of Electrical and Computer Engineering, Seoul National University, Seoul, Republic of Korea. [6]Bio-MAX Institute, Seoul National University, Seoul, Republic of Korea. [7]Zoonotic Virus Laboratory, Institut Pasteur Korea, Seongnam, Republic of Korea. [8]Department of Biomedical Sciences, Seoul National University College of Medicine, Seoul, Republic of Korea. [9]Department of Internal Medicine, Seoul National University College of Medicine, Seoul, Republic of Korea. [10]These authors contributed equally: Seoryeong Park, Jaewon Choi. ✉e-mail: seungtaek.kim@ip-korea.org; mdohmd@snu.ac.kr; skwon@snu.ac.kr; jjhchung@snu.ac.kr

(BCR) repertoires of BNT162b2 vaccinees and traced the development of Omicron variant-neutralizing antibodies.

## Results

### Vaccination and plasma antibody levels

We followed 41 healthcare workers from Seoul National University Hospital who received two doses of the BNT162b2 mRNA vaccine with a three-week interval and a third dose at approximately nine months after the first dose (Fig. 1). Peripheral blood samples were collected six times: once before the first dose, three times after each dose, and two times between the second and third doses. The vaccinees did not report a history of COVID-19 infection, and the absence of plasma IgG against the nucleocapsid (N) protein of SARS-CoV-2 was confirmed in all vaccinees (Supplementary Fig. 1). In an enzyme-linked immunosorbent assay (ELISA), plasma levels of IgA and IgG against the ancestral RBD were significantly increased after the second dose (Supplementary Fig. 2). However, when vaccination ELISA was performed against BA.1 RBD, a third dose was essential to achieve elevated levels of IgA and IgG, which also reacted to the RBD of Omicron subvariant BQ.1.1 (Supplementary Figs. 2 and 3). This observation was in accord with

prior reports that neutralizing activity of plasma against the Omicron variant was detectable only after the third dose[19,20], while the second dose could induce Omicron variant-neutralizing memory B cells[21].

### Selection and characterization of BA.1 RBD-reactive clones

We selected six vaccinees and generated a single-chain variable fragment (scFv) phage display library using cDNA prepared from the sixth blood sample. For the selection of these vaccinees, we reconstituted the in silico chronological BCR repertoires of vaccinees by next-generation sequencing and compared the frequency of BCR heavy chain (HC) clonotypes listed in the CoV-Ab Dab[22] database binding the SARS-CoV-2 spike protein. Vaccinee 32, 35, 39 and 43 and vaccine 22 and 27 showed a higher frequency of clonotypes binding to the BA.1 and ancestral spike proteins, respectively, and were selected. In this study, the BCR HC clonotype was defined as a set of immunoglobulin heavy chain sequences that exhibit the same immunoglobulin heavy variable (IGHV) and immunoglobulin heavy joining (IGHJ) genes and homologous heavy chain complementarity-determining region 3 (HCDR3) amino acid sequences with a minimum sequence identity of 80% to a reference sequence.

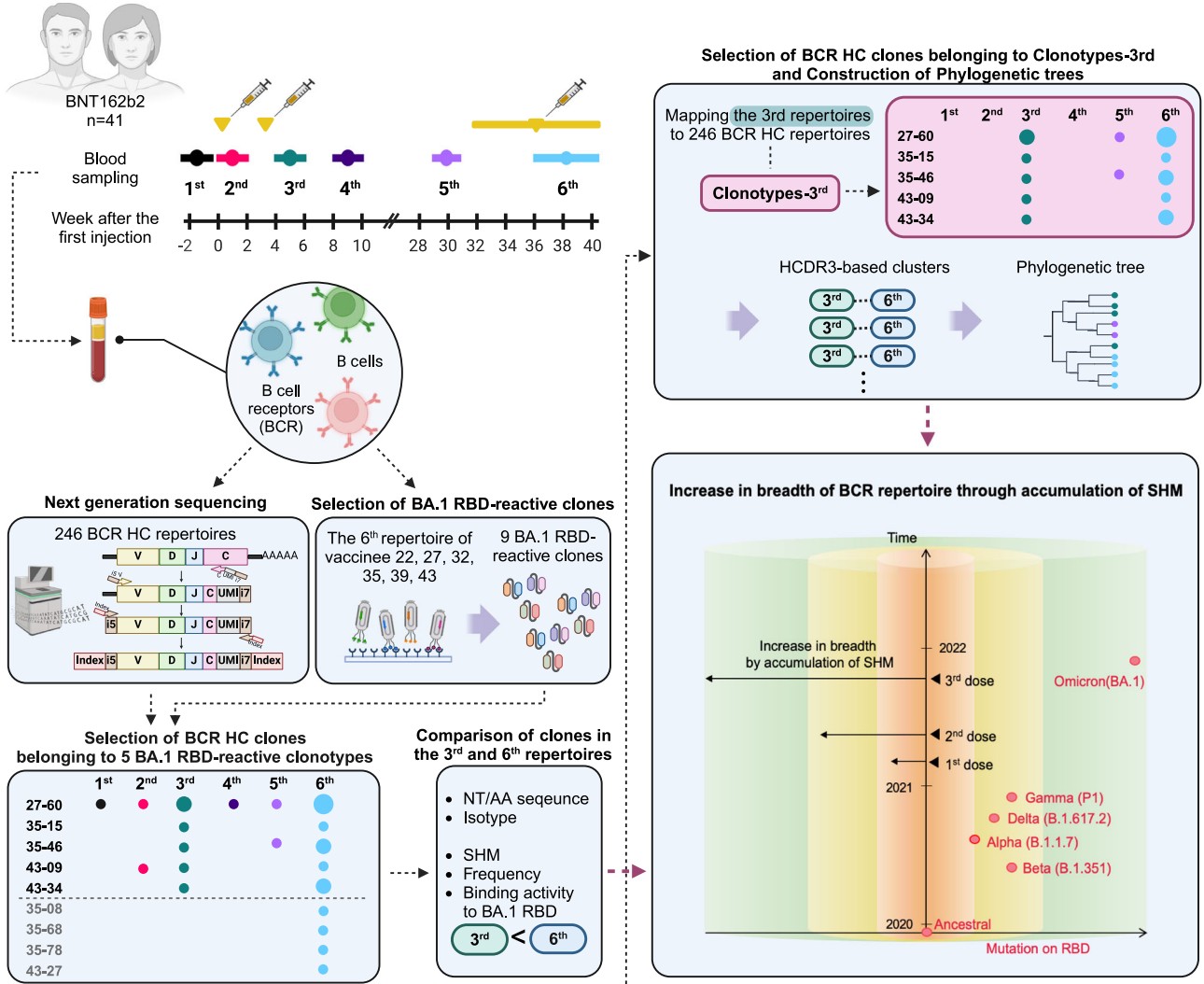

**Fig. 1 | A comprehensive schematic overview of the study.** Vaccinees received the first two doses at a three-week interval and the third dose between the 31st and 39th weeks after the first dose. Blood was collected six times: pre-vaccination (1st); 1 week after the first dose (2nd); 1, 6, and 30 weeks after the second dose (3rd, 4th, and 5th); and one to four weeks after the third dose (6th). The following illustration provides an overview of the overall study design and analysis. In the last box, the x-axis and y-axis represent mutations on the RBD and a timeline of COVID-19 variants and vaccinations, respectively. The cylindrical plot illustrates the breadth of the BCR repertoire following vaccination. The red dots represent SARS-CoV-2 variants. NT nucleotides, AA amino acids.

**Table 1 | BA.1 RBD-reactive scFv clones and their BCR-HC clonotypes**

| scFv clone | scFv library (vaccinees) | HCDR3 sequences | IGHV | IGHJ | No. of vaccinees with the corresponding BCR HC clonotype |
|---|---|---|---|---|---|
| 27-60 | 22, 27, 32, 39, 43 | ARDLMEAGGMDV | *IGHV3-53/3-66* | *IGHJ6* | 19 |
| 35-08 | 35 | AREIGYSGSGSAKYFDP | *IGHV1-69* | *IGHJ5* | 1 |
| 35-15 | 35 | ARAEDHGTYYSDSSGYHFDY | *IGHV1-69* | *IGHJ4* | 1 |
| 35-46 | 35 | ARVHGYSGYGANDAFDI | *IGHV1-69* | *IGHJ3* | 2 |
| 35-68 | 35 | AREVGYSGFGASPKFDP | *IGHV1-69* | *IGHJ5* | 1 |
| 35-78 | 35 | AREPGILGYCSSTSCYID | *IGHV1-69* | *IGHJ4* | 1 |
| 43-09 | 43 | ARTGSGWTDAFDI | *IGHV3-30* | *IGHJ3* | 5 |
| 43-27 | 43 | ATTYHYDTDGPYGEFYY | *IGHV5-51* | *IGHJ4* | 1 |
| 43-34 | 43 | ARVRGYSGYGASGYFDN | *IGHV1-69* | *IGHJ4* | 1 |

From six libraries, nine BA.1 RBD-reactive scFv clones were selected based on biopanning and phage ELISA. A cluster of scFv clones composed of 27–60 clones and 51 other clones sharing *IGHV3-53/3-66*, *IGHJ6*, and the same HCDR sequences at the amino acid level was selected from five libraries (Table 1 and Supplementary Data 1). We have previously reported that an *IGHV3-53/3-66* and *IGHJ6* BCR HC clonotype is present not only in the majority of convalescent patients from ancestral strains but also in normal human populations not exposed to SARS-CoV-2 and can neutralize the virus[23]. The frequent use of the *IGHV3-53/3-66* and *IGHJ6* gene pair in SARS-CoV-2 neutralizing antibodies was further confirmed by a subsequent report[24–26]. Six scFv clones carrying the *IGHV1-69* gene were selected from two libraries, which frequently encoded RBD-reactive antibodies found in convalescent patients from both the ancestral strain and BA.1 variant[27]. Two additional scFv clones were selected from the two libraries.

In the recombinant scFv-human Fc-hemagglutinin (HA) fusion protein format, seven scFv clones showed reactivity against the RBD of the ancestral virus, Alpha, Beta, Gamma, and Delta variants and Omicron-sublineage variants (BA.1, BA.2, BA.4/5 and BQ.1.1) (Supplementary Fig. 4a), with half-maximal effective concentrations (EC50) below 350 pM. The 27–60 clone and 35–15 clone showed dramatically reduced reactivity to BA.4/5 and BQ.1.1 RBDs and BQ.1.1 RBD, respectively. In a microneutralization assay, all clones neutralized the ancestral SARS-CoV-2 strain with a half-maximal inhibition concentration (IC50) range of 0.44 to 10.03 µg/ml (Supplementary Fig. 4b). Four clones (35-8, 35–46, 35–68, and 35–78) showed slightly decreased neutralizing activity against the BA.1 variant with an IC50 below 15 µg/ml, while the activity of others was increased to 25.79 to 74.12 µg/ml. Three clones (35-8, 35–68, and 35–78) could also neutralize Alpha, Beta, Gamma and Delta variants with less IC50 values under 12.12 ug/ml.

## BA.1 RBD-reactive clonotypes in BCR HC repertoires

Total 246 BCR HC libraries were used to construct BCR HC clonotypes (Supplementary Fig. 5). BCR HC clonotypes of BA.1 RBD-reactive clones (BA.1 RBD-reactive BCR HC clonotype) were mapped to 293 sequences in the repertoire of 23 vaccinees (Table 1 and Supplementary Data 2). The most frequently identified BCR HC clonotype was 27–60, found in 19 vaccinees (46%), followed clonotypes by 43-09 and 35–46, found in five and two vaccinees (12% and 5%), respectively (Supplementary Data 3). Six BCR HC clonotypes were found only in one vaccinee. Five BCR HC clonotypes, including 27–60, were not found in vaccinees in whom the antibody clones were found. This type of discrepancy is expected to inevitably originate from the incomplete in silico reconstitution of the BCR HC repertoire due to the limitation of the throughput of next-generation sequencing[28]. In this regard, the BCR HC clonotypes are expected to be present in a larger population than we observed herein. Also, as the IGHV-specific forward primers used in this study were not targeting the very 5' end of the IGHV gene, the probability of inaccurate allele annotation is approximately 12.96%.

However, we also believe that these limitations would minimally affect the interpretations and conclusions drawn from our results.

Among the 293 BCR HC sequences, 57 and 216 BCR HC sequences were present in the third and sixth repertoires, respectively. Only 20 BCR HC sequences were found in other chronological BCR repertoires. This skewed distribution originated from the expansion and proliferation of B cells encoding the BA.1 RBD-reactive BCR HC clonotype immediately after the second and third doses of the vaccine. Thus, we limited our statistical analysis to BCR HC sequences present in the third and sixth repertoires. Rapid class switch recombination (CSR) following vaccination was evident, as the main immunoglobulin subtype of BA.1 RBD-reactive BCR HC sequences was IgG$_1$ in the third repertoire, followed by IgG$_2$, which was maintained in the sixth repertoire (Supplementary Data 3 and Supplementary Fig. 6a). This rapid CSR resulting in the dominance of the IgG$_1$ subtype among RBD-reactive BCR HC sequences was also observed in convalescent patients infected with the ancestral virus[23].

Additionally, SHM of BA.1 RBD-reactive BCR HC clonotypes occurred mainly after the third dose. The average number of SHMs in the IGHV gene of BA.1 RBD-reactive BCR HC clonotypes in the sixth repertoire was dramatically increased compared to that in the third repertoire with statistical significance (Supplementary Fig. 6b). The diversity of the HCDR3 sequence also increased from the third repertoire to the sixth repertoire (Supplementary Fig. 6c). Five BA.1 RBD-reactive BCR HC clonotypes were mapped to eight vaccinees in both the third and sixth repertoires (Supplementary Fig. 6d). The increase in the average number of SHMs was evident, and statistical significance was confirmed in three vaccinees with the 27–60 clonotype and one vaccinee with the 43-34 clonotype.

## Development of concomitant reactivity to ancestral and BA.1 RBDs via SHM

Clone 27–60 was encoded by *IGHV3-53/IGHV3-66* (Fig. 2a), in which CDR1 and CDR2 are known to have two key motifs providing reactivity to ancestral RBD[23,29]. The BCR HC of clone 27–60 had two SHMs in its IGHV gene sequence, and when we back-mutated the V27 residue in CDR1 or the F58 residue adjacent to CDR2 to the germline sequence, its affinity for the ancestral and BA.1 RBDs was decreased 8.7-fold and greatly diminished, respectively (Fig. 2b). This observation showed that the reactivity of clone 27–60 toward the BA.1 RBD depended on the SHM to a greater degree than its reactivity to the ancestral RBD.

Thereafter, we selected the most frequent BCR HC sequences of the 27–60 clonotype in the third and sixth BCR repertoires of six vaccinees and analyzed their sequences (Fig. 2a). All the BCR HC sequences found in the sixth repertoires had more SHMs than their pairs in the third repertoires. All six BCR HC sequences in the sixth repertoire gained the Y58F mutation, increasing the affinity of *IGHV3-53/IGHV3-66* antibodies for the ancestral RBD[30,31]. There was also diversity in their HCDR3 sequences limited to a motif of three amino acid residues. As the HCDR3 sequence can diversify either at the stage

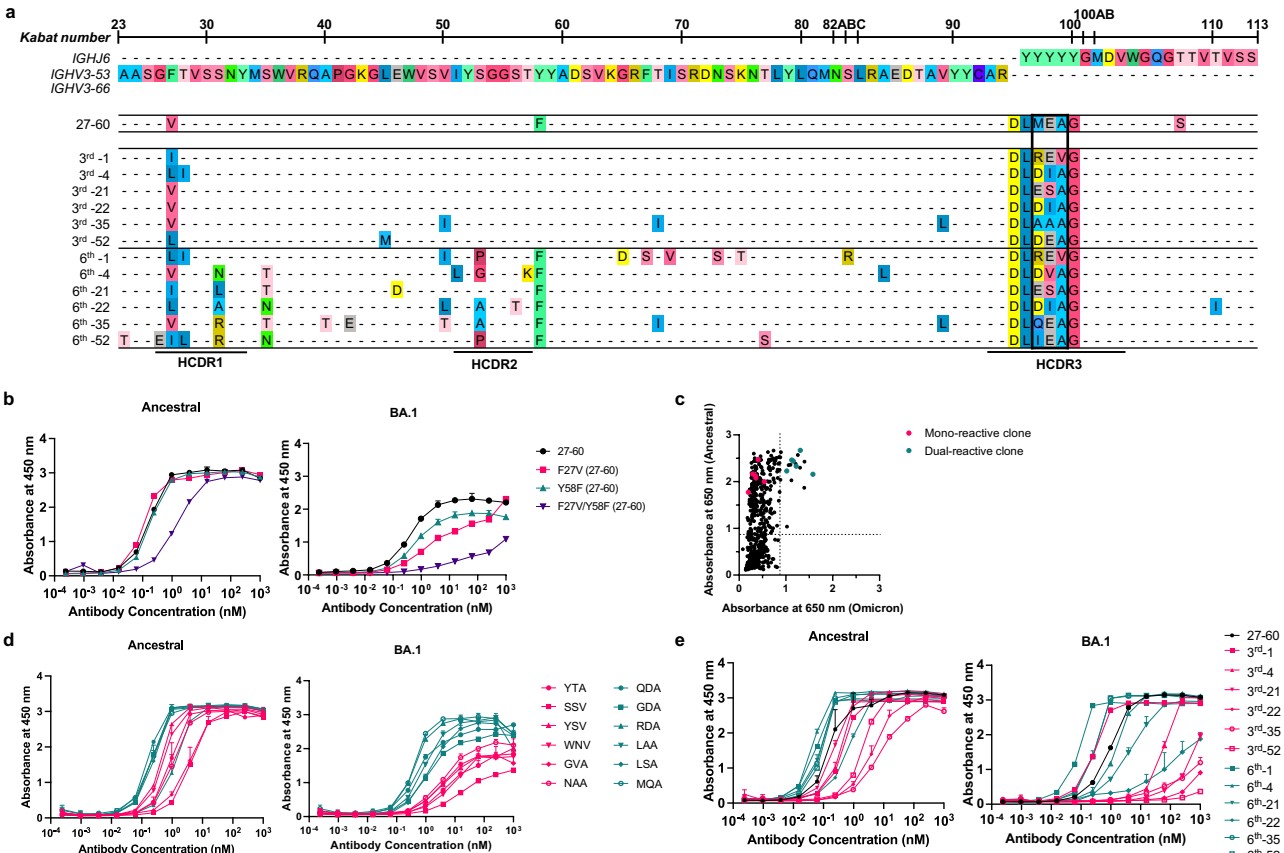

**Fig. 2 | Characterization of the 27–60 BCR HC clonotype. a** BCR HC sequences found in six vaccinees in their third and sixth BCR repertoires with the highest frequency. **b** Reactivity of the recombinant 27–60 scFv-hFc-HA protein and its clones back mutated to the germline sequence. **c** Reactivity of scFv-displaying phage clones from the HCDR3-randomized library to the ancestral and BA.1 RBDs. The clones labeled in red and green were selected for expression as recombinant scFv-hFc-HA protein. **d** Reactivity of recombinant scFv fusion proteins of HCDR3-randomized (ARDLXX(A/V)GGMDV) clones to the ancestral and BA.1 RBDs. **e** Reactivity of recombinant scFv-hFc-HA proteins encoded by individual BCR HC sequences and the light chain gene of clone 27–60 to the ancestral and BA.1 RBDs. All experiments were performed in duplicate, and the data are presented as the means ± SD. Source data are provided as a Source Data file.

of germline VDJ gene recombination by P- and N-nucleotide addition/deletion or later by SHM, it was not possible to determine the germline HCDR3 sequence of clones 27–60. To analyze the influence of the HCDR3 sequence on reactivity to ancestral and BA.1 RBDs, we generated a scFv phage display with the randomized HCDR3 sequence of ARDLXX(A/V)GGMDV, arbitrarily selected 856 phage clones and checked their reactivity to both RBDs. For the ancestral RBD, 267 phage clones (31%) showed positive reactivity (Fig. 2c and Supplementary Fig. 7). However, for the BA.1 RBD, only 18 phage clones (2%) were reactive. All 18 clones were dual-reactive to both RBDs. Six mono-reactive and six dual-reactive phage clones were arbitrarily selected for expression as recombinant scFv-hFc-HA fusion proteins and tested for their affinity for both RBDs. The EC50 values of mono-reactive scFv proteins to the ancestral and BA.1 RBDs were in the ranges of 0.84–3.80 nM and 3.83–17.25 nM, respectively. In addition, dual-reactive scFv proteins showed EC50 values of 0.16–1.29 nM and 0.35–1.26 nM for the ancestral and BA.1 RBDs, respectively (Fig. 2d). These observations proved that it is relatively rare for ancestral RBD-reactive BCR HC clonotypes encoded by the *IGHV3-53/3-66* and *IGHJ6* genes with the HCDR3 sequence of ARDLXX(A/V)GGMDV (BCR HC XXA/V clonotypes) to develop increased compatible reactivity to the BA.1 RBD. This dual reactivity is likely to be achieved among the large population of antigen-stimulated and proliferating B cells through SHM, rather than in a small number of B cells in the stage of germline VDJ gene rearrangement. Indeed, only one BCR HC XXA/V sequence was found in only one vaccinee in the pre-immune repertoire

(Supplementary Fig. 8). After the first, second, and third injections, 3, 58, and 138 BCR HC XXA/V sequences were found in two, ten and sixteen vaccinees, respectively. The average frequency of BCR HC XXA/V sequences also increased from $6.06 \times 10^{-6}$ under pre-immune status to $2.85 \times 10^{-5}$, $1.91 \times 10^{-5}$, and $2.41 \times 10^{-5}$ after the first, second, and third doses, respectively.

To check the effect of SHM accumulation on the affinity for RBDs in 27-60 BCR HC clonotypes, six pairs of BCR HC sequences were coupled with the light chain gene of clone 27–60 (Supplementary Data 4) and expressed as recombinant scFv-hFc-HA fusion proteins. Then, their affinity for the ancestral RBD and BA.1 RBD was determined by ELISA. For the ancestral RBD, all six scFvs of the third repertoire showed notable affinities, with EC50 values less than 7.5 nM. However, only one scFv from vaccinee 1 showed comparable affinity for BA.1 RBD (Fig. 2e). In the sixth repertoire, four additional scFvs showed affinity for the BA.1 RBD, with EC50 values less than 4.5 nM.

We also analyzed the BCR HC sequences of four additional clonotypes (35-15, 35–46, 43-09 and 43–34) found in both the third and sixth repertoires of identical vaccinees (Supplementary Fig. 9a). All BCR HC sequences in the sixth repertoires showed more SHMs than their paired sequences in the third repertoires. Then, we prepared eight recombinant scFv-hFc-HA fusion proteins in pairs with the light chains of four scFv phage clones. In ELISA, all scFvs from the third repertoire showed a remarkable level of affinity for the ancestral RBD, with EC50 values less than 200 pM, which were well maintained within a 2-fold difference in their paired sequences in the sixth repertoire

(Supplementary Fig. 9b). It was exceptional that the BCR HC sequences of the 35–46 clonotype in the third library were encoded by *IGHV1-69/IGHJ3* genes without any SHM and showed a significantly high affinity for BA.1 RBD. In the 35-15, 43-09 and 43-34 BCR HC clonotype pairs, the affinity for BA.1 RBD was significantly increased from 0.59, 1.12 and 7.91 nM in the third repertoire to 0.13, 0.09 and 0.14 nM in the sixth repertoire, respectively.

## Expansion of the BCR HC repertoire to RBDs of Omicron subvariants by SHM

We studied how the diversification of the BCR HC repertoire by SHM expands its reactivity to RBDs of Omicron subvariants. We first searched the BCR HC sequences of the 27–60, 35-15, 35–46, 43-09 and 43-34 clonotypes in the third repertoire and found them in nine, one, one, one, and one of the vaccinees, respectively (Supplementary Data 3). Then, we selected the BCR HC sequences with the highest frequency in each vaccinee and used them to construct a new set of BCR HC clonotypes (designated clonotypes-3rd) confined to each vaccinee. The greatest numbers of BCR HC sequences in the 27–60, 35-15, 35–46, 43-09 and 43-34 clonotypes-3rd were found in vaccinees 35, 43, 35, 23, and 43, consisting of 86, 8, 24, 2, and 18 sequences, respectively. Based on their HCDR3 sequences, these sequences in the clonotypes-3rd were subdivided into 11, 4, 11, 2, and 9 clusters, respectively (Fig. 3). From each cluster, the BCR HC sequences with the highest frequency that were present in the third and sixth BCR HC repertoires were selected, and their genes were synthesized (Supplementary Fig. 10, 11 and Supplementary Data 4). As expected, all the scFv clones in the sixth BCR HC repertoires showed more SHM than those in the third repertoires (Fig. 3), and some of them showed dramatically enhanced binding toward specific types of Omicron variant RBDs. In the sixth repertoires, four, three, seven and seven clones showed potent reactivity to BA.1, BQ.1.1, XBB.1.5, and XBB.1.16 RBD with an EC50 below 0.100 nM, respectively. These findings suggest that the accumulation of SHM after the third dose of ancestral SARS-CoV-2 vaccine generated clones with divergent mosaic pattern of specificity toward four major Omicron subvariants. More interestingly, clones reactive to BQ.1.1, XBB.1.5, and XBB.1.16 achieved their specificity much earlier than their emergence.

Our study showed that the third dose of the mRNA vaccine encoding the ancestral viral spike protein induced the accumulation of SHM in ancestral RBD-reactive antibody clones and generated a panel of BCRs with additional divergent reactivity to RBDs of Omicron subvariants, which may contribute to the dramatically increased plasma level of BA.1 RBD–reactive antibodies and may have the potential to counteract novel SARS-CoV-2 variants yet to emerge (Fig. 1). We believe that this expansion in the specificity of the BCR repertoire to unseen antigens via SHM is a protective immune mechanism that evolved in response to the challenge of viral immune escape.

## Methods
### Study participants
Peripheral blood sampling was approved by the Institutional Ethics Review Board of Seoul National University Hospital (IRB approval number, 2102-032-1193). Informed consent was obtained from all participants of this study. The vaccinees had a median age of 30 years (range 23–62) and showed a nearly equal distribution of males and females (46% and 54%, respectively). Demographic data for the 41 vaccinees are summarized in Supplementary Data 5. Peripheral blood mononuclear cells (PBMCs) and plasma were separated using Lymphoprep (STEMCELL) Ficoll (Cytiva) following the manufacturer's instructions. Total RNA was isolated using TRIzol Reagent (Invitrogen) according to the manufacturer's instructions.

### ELISA
Plasma and phage ELISAs were performed in 96-well microtiter plates (Corning, 3690) coated with 100 ng of recombinant SARS-CoV-2 proteins (Sino Biological, Ancestral RBD, 40592-V08H; Alpha RBD, 40592-V08H82; Beta RBD, 40592-V08H85; Gamma RBD, 40592-V08H86; Delta RBD, 40592-V08H90; Omicron BA.1 RBD, 40592-V08H121; Omicron BA.2 RBD, 40592-V08H123, Omicron BA.4/5 RBD, 40592-V08H130; Omicron BQ.1.1 RBD, 40592-V08H143; Omicron XBB.1.5, 40592-V08H146; Omicron XBB.1.16, 40592-V08H136; RBD Ancestral N, 40588-V08B) in coating buffer (0.1 M sodium bicarbonate, pH 8.6) as described previously[23]. Briefly, the plates were coated with the antigen by incubation at 4 °C overnight and blocked with 3% bovine serum albumin (BSA) in PBS for 1 h at 37 °C. Then, serially diluted plasma of 41 vaccinees and one COVID-19 patient[23] (100-, 500-, 2,500-fold) or phage supernatant (twofold) in blocking buffer was added to the wells of microtiter plates, followed by incubation for 1 h at 37 °C. Then, the plates were washed three times with 0.05% PBST. Horseradish peroxidase (HRP)-conjugated goat anti-human IgM and IgA (Invitrogen, A18835 and A18781, 1:5,000), rabbit anti-human IgG antibody (Invitrogen, 31423, 1:20,000) and HRP-conjugated anti-M13 antibody (Sino Biological, 11973-MM05T-H, 1:4,000) were used to determine the amount of bound antibody or M13 bacteriophage. A 3,3′,5,5′-tetramethylbenzidine liquid substrate solution (Thermo Fisher Scientific Inc.) was used as an HRP substrate.

In ELISA for the recombinant scFv-hFc-HA fusion proteins, the wells of microtiter plates (Corning) were first coated with 100 ng of mouse anti-His antibody (Invitrogen, MA1-21315) and blocked. Then, the recombinant SARS-CoV-2 RBD protein with a polyhistidine tag (100 nM) was added to the wells. After brief washing, the scFv-hFC-HA fusion proteins were serially diluted fourfold from 1 µM to 0.24 pM in blocking buffer and added to wells of a microtiter plate. HRP-conjugated rat anti-HA antibody (Roche, 12013819001, 1:1,000) was used to determine the amount of bound antibody. All assays were performed in duplicate. The absorbance was measured at either 450 or 650 nm using a microplate spectrophotometer (Thermo Fisher Scientific Inc., Multiskan GO), depending on the use of 2 M sulfuric acid as the stop solution. All ELISA data were analyzed using GraphPad Prism software v6 (GraphPad Software).

### Next-generation sequencing
Genes encoding the variable domain of the heavy chain ($V_H$) and part of the first constant domain of the heavy chain (CH1) domain ($V_H$-CH1) were amplified using specific primers, as described previously[23]. All primers used are listed in Supplementary Data 6. These six IGHV-specific primers are expected to amplify a total of 269 out of 270 IGHV alleles (99.63%) available in the IMGT database. *IGHV4-39*08* is expected to encounter difficulty in amplification, due to a mismatch with the germline sequence at the third nucleotide from the 3′ end of the primer[32] (Supplementary Data 7). Briefly, total RNA was used as a template to synthesize complementary DNA (cDNA) using the SuperScript IV First-Strand Synthesis System (Invitrogen) with specific primers (primer No. 1–8) targeting the CH1 domain of each isotype (IgM, IgD, IgG, IgA, and IgE) according to the manufacturer's protocol. After cDNA synthesis, 1.8 volumes of SPRI beads (Beckman Coulter, AMPure XP) were used to purify cDNA, which was eluted in 35 µl of water. The purified cDNA (15 µl) was subjected to second-strand synthesis in a 25 µl reaction volume using IGHV gene–specific primers (primer No.9–14) and a KAPA Biosystems kit (Roche, KAPA HiFi HotStart). The PCR conditions were as follows: 95 °C for 3 min, 98 °C for 30 s, 60 °C for 45 s, and 72 °C for 6 min. After second-strand synthesis, dsDNA was purified using 1 volume of SPRI beads, as described above. $V_H$-CH1 genes were amplified using purified dsDNA (15 µl) in a 25 µl reaction volume using primers containing indexing sequences (primers 15 and 16) and a KAPA Biosystems kit. The PCR conditions were as follows: 95 °C for 3 min; 25 cycles of 98 °C for 30 s, 60 °C for 30 s, and 72 °C for 1 min; and 72 °C for 5 min. PCR products were subjected to electrophoresis on a 1.5% agarose gel and purified using a QIAquick gel extraction kit (QIAGEN Inc.) according to the manufacturer's

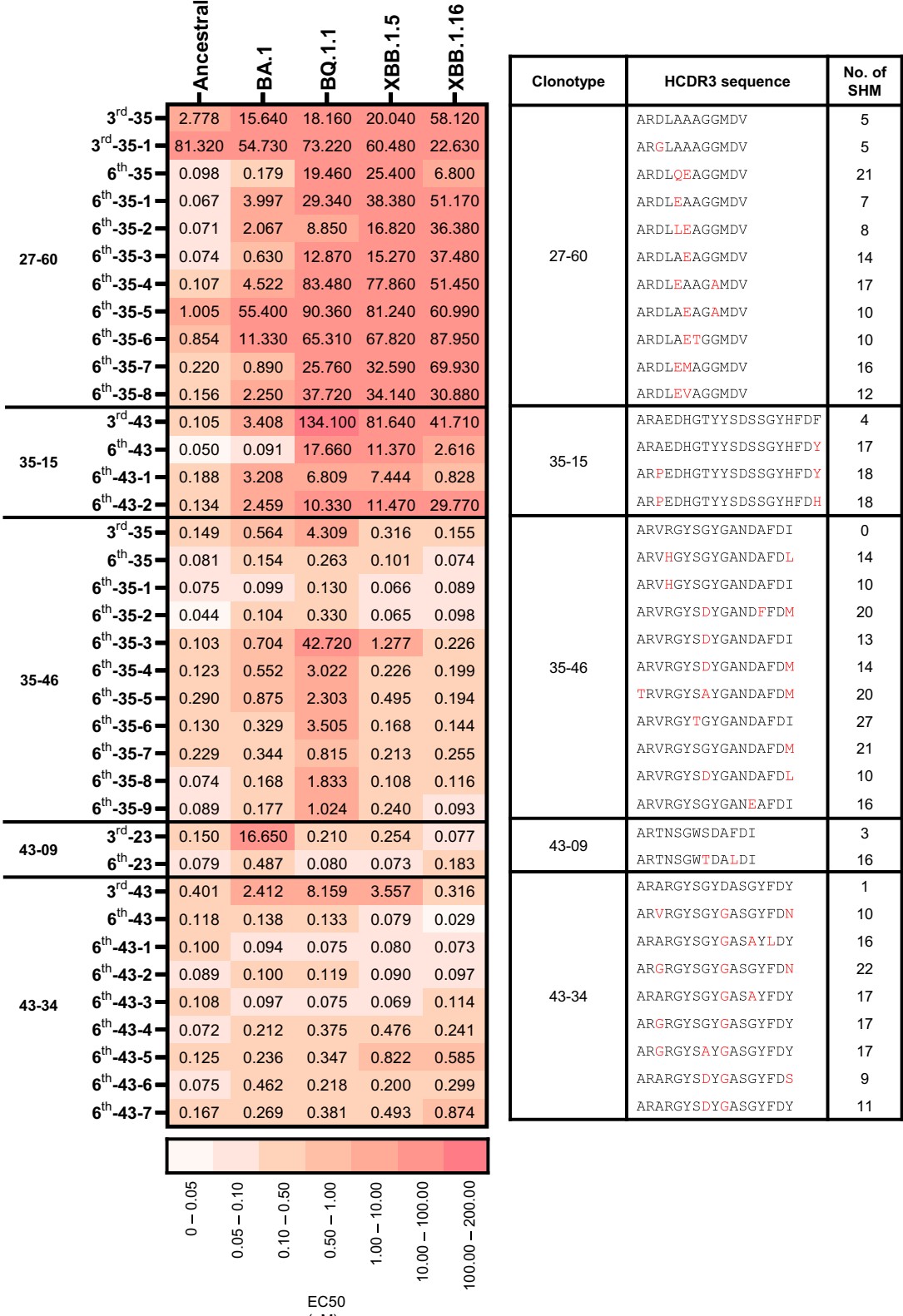

**Fig. 3 | The reactivity of the scFv clones of HCDR3-based clusters in the 27–60, 35-15, 35–46, 43-09, and 43-34 BCR HC clonotypes-3rd.** The EC50 values to Ancestral, BA.1, BQ.1.1, XBB.1.5, and XBB.1.16 RBDs of the scFv clones with the highest frequencies of HCDR3-based clusters in each clonotype-3rd are displayed with their HCDR3 sequences and the quantities of SHM.

instructions. The gel-purified PCR products were purified again using 1 volume of SPRI beads and eluted in 20 μl water. The SPRI-purified sequencing libraries were quantified with a 4200 TapeStation System (Agilent Technologies) using a D1000 ScreenTape assay and subjected

to next-generation sequencing on the Illumina NovaSeq 6000 250PE (SP Chip) platform using Novaseq 6000 reagent kits (Illumina Inc.). The total read counts of the chronological repertoires after NGS data processing is summarized in Supplementary Data 8.

## NGS data processing

Raw sequencing reads obtained from sequencing the $V_H$-CH1 region of B cells in peripheral blood were processed using a custom pipeline. The pipeline included adapter trimming and quality filtering; unique molecular identifier (UMI) processing; V(D)J gene annotation; clustering; quality control; and diversity analysis.

The forward reads (R1) and reverse reads (R2) of the raw NGS data were merged using paired-end read merger (PEAR) v0.9.10 with the default settings[33]. The merged reads were q-filtered under the q20p95 condition, resulting in 95% of base pairs in the reads having a Phred score greater than 20. Primer positions were identified in the quality-filtered reads, and primer regions were trimmed to remove the effects of primer synthesis errors while allowing one substitution or deletion. We identified 187 out of 270 IGHV alleles (69.3%) from the IMGT database across six time points in 41 vaccinees (246 BCR HC libraries) of Korean nationality. This ratio highly resembles the average allele distribution observed among Eastern Asian populations (62.7%)[34] (Supplementary Data 9).

Based on the primer recognition results, UMI sequences were extracted, and reads were clustered according to the UMI sequences. To eliminate index misassignment, we subclustered the clustered reads based on the similarity of the reads (allowing 5 mismatches in each subcluster) and matched the majority subcluster to the UMI. The subclustered reads were aligned using the multiple sequence alignment tool Clustal Omega v1.2.4 with the default settings[35,36]. Consensus calling was performed by selecting major frequency bases at every position of the aligned sequences. The number of reads in the consensus sequence was redefined as the number of UMI subclusters belonging to the consensus sequence.

Sequence annotation consisted of isotype annotation and V(D)J annotation. The consensus sequence was divided into a V(D)J region and a constant region. The isotype of the consensus sequence was annotated by aligning the extracted constant region with the constant gene of the International Immunogenetics Information System (IMGT)[37]. Then, the V(D)J region of the consensus sequence was annotated using an updated version of IgBLAST (v1.17.1)[38]. Among the annotation results, the IGHV genes, IGHJ genes, HCDR3 sequences, and number of SHMs were extracted for further analysis. The number of SHMs cannot be determined for sequences outside the primer binding sites; therefore, SHMs were calculated by comparing the amplified regions from primer-mediated amplification with the IGHV gene germline sequence. The nonfunctional consensus reads were defined and filtered using the following criteria: (i) sequence length shorter than 250 base pairs from each R1 and R2, (ii) presence of a stop codon or frameshift in the entire amino acid sequence, (iii) failure to annotate one or more HCDR1, HCDR2 and HCDR3 regions, and (iv) failure to annotate the isotype.

## Construction of scFv phage display library

Six human scFv phage display libraries were constructed using the total RNA prepared from the sixth blood sample for vaccine Nos. 22, 27, 32, 35, 39 and 43 as described previously[23]. Briefly, for the $V_H$ and $V_\kappa/V_\lambda$ genes, total RNA was employed to synthesize cDNA using the SuperScript IV First-Strand Synthesis System (Invitrogen) with gene-specific primers targeting the $J_H$ and $C_{\kappa/\lambda}$ genes (primer No.17–22), respectively. After cDNA synthesis, 1.8 volumes of SPRI beads (Beckman Coulter) were used to purify cDNA, which was eluted in 20 µl of water. The purified cDNA (11.25 µl) of the $V_H$ gene was subjected to second-strand synthesis in a 25 µl reaction volume using IGHV gene-specific primers (primer No. 23–32, 7.5 µl) and a KAPA Biosystems kit (Roche). The reaction conditions were as follows: 95 °C for 3 min, 98 °C for 1 min, 60 °C for 1 min, and 72 °C for 5 min. In the case of the $V_\kappa/V_\lambda$ gene, the eluted cDNA (17.25 µl) was used for the first round of PCR synthesis in a 25 µl reaction volume using $V_\kappa/V_\lambda$ and $J_\kappa/J_\lambda$ gene-specific primers (primer No. 33–69, 0.75 µl). The PCR conditions were as

follows: 95 °C for 3 min; 4 cycles of 98 °C for 1 min, 60 °C for 1 min, and 72 °C for 1 min; and 72 °C for 10 min. After second-strand synthesis for the $V_H$ gene or PCR amplification of the $V_\kappa/V_\lambda$ gene, double-stranded DNA (dsDNA) was purified with 1 volume of SPRI beads and eluted in 40 µl of water. $V_H$ and $V_\kappa/V_\lambda$ genes were amplified using 10 µl of purified dsDNA, 2.5 pmol of the primers (primer No. 70–73), and KAPA Biosystems kit components in a 50 µl total reaction volume with the following thermal cycling program: 95 °C for 3 min; 30 cycles of 98 °C for 30 s, 60 °C for 30 s, and 72 °C for 1 min; and 72 °C for 10 min. Then, the amplified $V_H$ and $V_\kappa/V_\lambda$ genes were subjected to electrophoresis on a 1.5% agarose gel and purified using a QIAquick gel extraction kit (QIAGEN Inc.) according to the manufacturer's instructions. The purified $V_H$ and $V_\kappa/V_\lambda$ gene fragments (100 ng) were mixed and subjected to overlap extension PCR to generate scFv genes using 2.5 pmol of the overlap extension primers (primer No. 74 and 75) using the KAPA Biosystems kit. The PCR conditions were as follows: 95 °C for 3 min; 25 cycles of 98 °C for 20 s, 65 °C for 15 s, and 72 °C for 1 min; and 72 °C for 10 min. The amplified scFv gene was purified and cloned into a phagemid vector[39].

## Selection of BA.1 RBD-reactive clones

Human scFv phage display libraries with $7.1 \times 10^8$, $6.1 \times 10^8$, $8.1 \times 10^8$, $8.3 \times 10^8$, $7.9 \times 10^8$, and $7.4 \times 10^8$ colony-forming units were generated using cDNA prepared from vaccinee Nos. 22, 27, 32, 35, 39 and 43, respectively. The libraries were subjected to five rounds of biopanning against the recombinant SARS-CoV-2 BA.1 RBD protein (Sino Biological Inc.) as described previously[40]. Immune tubes (SPL, 43015) coated with 17 µg of the BA.1 RBD were used for the first round, and $5 \times 10^6$ magnetic beads (Invitrogen, Dynabeads M-270 epoxy) conjugated with 1.4 µg of the BA.1 RBD protein were used for the other rounds. After each round of biopanning, the bound phages were eluted and amplified for the next round of biopanning. For the selection of BA.1 RBD-reactive scFv phage clones, individual phage clones were amplified from the titration plate of the last round and subjected to phage ELISA[41]. The genes encoding BA.1 RBD-reactive scFv clones were identified using phagemid DNA prepared from phage clones and Sanger nucleotide sequencing[40]. A recombinant scFv protein fused with human IgG1 FC and the HA peptide (scFv-hFc-HA) was expressed using a mammalian expression system with Expi293F™ cells (Gibco, A14527) and purified as described previously[41].

## HCDR3-randomized scFv phage display library

The gene fragment encoding the $V_H$ region with a randomized HCDR3 sequence and another gene fragment encoding the rest of the scFv were amplified using phagemid DNA of clone 27–60, primers for randomization (Primer No. 76–79) and the KAPA Biosystems kit. The PCR conditions were as follows: 95 °C for 3 min; 25 cycles of 98 °C for 20 s, 65 °C for 15 s, and 72 °C for 1 min; and 72 °C for 10 min. Then, the amplified genes were subjected to electrophoresis on a 1.5% agarose gel and purified using a QIAquick gel extraction kit (QIAGEN Inc.) according to the manufacturer's instructions. The purified gene fragments (200 ng) were mixed and subjected to overlap extension PCR to generate scFv genes using 2.5 pmol of the overlap extension primers (primer No.76 and 79) and the KAPA Biosystems kit. The PCR conditions were as follows: 95 °C for 3 min; 25 cycles of 98 °C for 20 s, 65 °C for 15 s, and 72 °C for 1 min; and 72 °C for 5 min. The amplified scFv genes were purified and cloned into a phagemid vector[38]. For phage ELISA, individual phage clones were amplified from the titration plate and subjected to phage ELISA[42].

## Microneutralization assay

The ancestral SARS-CoV-2 (βCoV/Korea/KCDC03/2020 NCCP43326), Alpha B.1.1.7 (hCoV-19/Korea/KDCA51463/2021 (NCCP 43381), Beta B.1.351 (hCoV-19/Korea/KDCA55905/2021 (NCCP 43382), Gamma P.1 (hCoV-19/Korea/KDCA95637/2021 (NCCP 43388), Delta B.1.617.2

(hCoV-19/Korea/KDCA119861/2021 (NCCP 43390), and Omicron B.1.1.529 (hCoV-19/Korea/KDCA447321/2021 NCCP43408) viruses were obtained from the Korea Disease Control and Prevention Agency. The viruses were propagated in Vero cells (ATCC, CCL-81) in Dulbecco's Modified Eagle's Medium (DMEM, Welgene) in the presence of 2% fetal bovine serum (Gibco, Thermo Fisher Scientific Inc.)[23,43]. Neutralization assays were performed as described previously[23,44]. Briefly, Vero cells were seeded in 96-well plates ($1.5 \times 10^4$ or $0.5 \times 10^4$ cells/well) in Opti-PRO SFM (Thermo Fisher Scientific Inc.) supplemented with 4 mM L-glutamine and 1× Antibiotics-Antimycotic (Thermo Fisher Scientific Inc/) and grown for 24 h at 37 °C in a 5% $CO_2$ environment. Recombinant scFv-hFc proteins were diluted from 100 to 0.1953 μg/ml (twofold) in phosphate-buffered saline (PBS, Welgene) and mixed with 100 or 500 $TCID_{50}$ of SARS-CoV-2. Then, the mixture was incubated for 30 min at 37 °C and added to the cells in tetrads, followed by incubation for 4 or 6 days at 37 °C in a 5% $CO_2$ environment. The cytopathic effect (CPE) in each well was visualized following crystal violet staining 4 or 6 days post infection, and measured using the EVOS Digital Inverted Imaging System with 40X lens (AMG). The $IC_{50}$ values were calculated using the dose–response inhibition equation of GraphPad Prism v6 (GraphPad Software).

### Construction of BCR HC clonotype-3[rd]

The most frequent BCR HC sequences of the 27–60, 35-15, 35–46, 43-09, and 43-34 clonotypes in the third BCR repertoires were mapped to the chronological BCR repertoires following the same definition of BCR HC clonotypes. All the mapped nucleotide sequences were aligned using the multiple sequence alignment tool Clustal Omega v1.2.4 with the default settings[35,36] and processed using Ugene software v1.16.2. The aligned mapped nucleotide sequences were interpreted using a phylogenetic tree generated by IgPhyML v1.1.3 052020 using the HLP model option[45]. The phylogenetic trees were plotted using Interactive Tree of Life (iTOL) online tool v6[46].

### Reporting summary

Further information on research design is available in the Nature Portfolio Reporting Summary linked to this article.

## Data availability

All sequencing data are available from the National Center for Biotechnology Information (www.ncbi.nlm.nih.gov/) under accession number PRJNA945512 (SRA). Source data are provided with this paper.

## Code availability

NGS data preprocessing was performed as described in a previous study (Kim, S. I. et al. Sci Transl Med 13[23]). All custom scripts are available on GitHub (https://github.com/BiNEL-SNU/SARS-CoV-2).

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

## Acknowledgements

This research was supported by the Korea Dementia Research Center (KDRC, HU20C0339, J. Chung), Korea Health Technology R&D Project through the Korea Health Industry Development Institute (KHIDI, HI23C0521, J. Chung), the BK21 FOUR program (J. Chung and S. Kwon), the National Research Foundation of Korea (NRF-2020R1A3B3079653 NRF-2020M3H1A1073304, NRF-2017M3A9G6068245, NRF-2022M3A9J1081343, NRF-2023M3A9G6057281, and RS-2024-00398073, S.T.K.), the SNUH Research Fund (03-2021-0080, W.P.), and the Inter-university Semiconductor Research Center (W.P.). The pathogen resources (NCCP43326, NCCP43381, NCCP43382, NCCP43388, NCCP43390, NCCP43408) for this study were provided by the National Culture Collection for Pathogens. J. Choi is grateful for financial support from the Hyundai Motor Chung Mong-Koo Foundation. Figure 1 was with BioRender.com.

## Author contributions

S.P. designed and conducted all experiments, performed analyses, interpreted experimental results, and wrote and revised the paper. J. Choi performed the bioinformatic analyses, visualized and interpreted the results, and wrote and revised the paper. Y.L., J.N. and N.P.K. performed the bioinformatic analysis. J.L. conducted experiments related to the microneutralization assay. D.Y. conducted experiments related to the construction of repertoire libraries for NGS. G.C. and S.J.K. contributed to experiments related to the expression of recombinant proteins. C.K., P.C., N.J.K. and W.P. enrolled the study participants and collected the study samples. M.O. conceived and designed the study, and interpreted all results. S.T.K. supervised all the experiments with infectious viruses and microneutralization assay. S. Kwon conceived the study and designed and supervised the bioinformatics analysis. J. Chung conceived the study, designed and supervised all experiments, interpreted all results, and wrote the paper.

## Competing interests

The authors declare no competing interests.
