## [Peer Review File · Nature Communications]

An ancestral SARS-CoV-2 vaccine induces anti-Omicron variants antibodies by hypermutationReviewers' Comments:

Reviewer #1:

Remarks to the Author:

The manuscript by Park and colleagues aims to associate SHM accumulation with a mechanism against virus variant immune escape.

It is potentially a good idea and good dataset but it needs substantial redesign and improvement on data visualization to support the claims.

Furthermore, neutralization assays with scFv might not inform the effector function of antibodies, thus, cannot totally inform immune escape as claimed by the authors.

Reviewer #2:

Remarks to the Author:

The manuscript by Park et al. entitled "An ancestral SARS-CoV-2 vaccine induces anti-omicron variants antibodies by hypermutation" provide a clear and coherent analysis of the B-cell receptor (BCR) repertoires in patients receiving multiple doses of the original vaccine BNT162b2 against the SARS-CoV-2. The authors convincingly show that after a third boost of the BNT162b2 vaccine a significant part of the subjects also produce neutralizing antibodies against the omicron variant. Based on the data presented the authors suggest that the neutralizing antibodies against the omicron variant is most like due to hypermutations and less like due to VDJ recombination.

The study is well performed and provide significant novelty to our present day understanding of how protective immunity arises.

A number of minor details could be improved as indicated below, but otherwise I would recommend that the manuscript is accepted after minor revisions.

- In line 56-57 the authors write that the inefficiency of the vaccines has led to reduced efficacy of most commercially available monoclonal antibodies and antibodies under development against BA.1. In connection with the description with reduced efficiency of the BNT162b2 vaccine, I do not understand why the authors suddenly comment on the therapeutic antibodies under development.
- The authors test all the subjects being vaccinated for whether they have had previous exposure to SARS-CoV-2 by testing whether antibodies recognizing the N-protein from SARS-CoV-2. While this is a valid approach, the data presented in extended data figure 1 lack a control. Sera from patients testing positive for SARS-CoV-2 should have been included.
- In line 124 the author writes that five BCR HC clonotypes, including the 27-60, were not found in vaccinees in who the antibody clones were found. This is suggested to be due to technical issues with the Next generation sequencing. It would be good if the authors comments on how such issues may affect the general conclusions from their study.
- In line 168 and forward, the authors describe the construction of a scFv phage display library, where they arbitrarily selected 672 phage clones and tested their reactivity to both RBDs. The library contained randomized residues and a residue varied between A and V, thus the library should contain around 800 different phage antibody clones. To what extend does the 672 selected phage clones represent different antibodies?
- In line 169 the author describes that 30 % of the selected phage antibodies bind the ancestral RBD but only 18 the BA.1 RBD – could the authors comment on this taking into account the my comment made above.

Reviewer #3:

Remarks to the Author:

This study addresses an interesting question, namely why a third dose of the BNT162b2 covid-19 mRNA vaccine, which encodes the ancestral SARS-CoV-2 spike protein, stimulates the production of antibodies that are capable of neutralizing Omicron subvariants. This is a well-known observation and

the authors cite references 14, 15, 16, 17 and 18 to this effect. They state that the mechanism by which Omicron-neutralizing antibodies are generated following repeated vaccinations with the ancestral spike remains unclear. However, this is not quite true as there are previous publications on this topic, which the authors should include in their reference list (along with qualifying their statement) such as these:

Rapid Hypermutation B Cell Trajectory Recruits Previously Primed B Cells Upon Third SARS-CoV-2 mRNA Vaccination - PubMed (nih.gov)

Cross neutralization of SARS-CoV-2 omicron subvariants after repeated doses of COVID-19 mRNA vaccines - PubMed (nih.gov)

The authors collect blood from different time points following vaccination and show plasma ELISA results in Figure 1. This type of data has been shown in several previous papers and should be moved to supplementary materials. They then selected six vaccine recipients that were vaccinated three times with the BNT162b2 mRNA vaccine and generated single-chain variable fragment (scFv) phage display libraries that were sequenced by Next Generation Sequencing on the Illumina NovaSeq platform. They use CoV-Ab Dab and panning to identify spike-specific antibody clonotypes in the B cell receptor repertoire data. They focus on 9 clonotypes, all of which were similar to Ab clonotypes that were previously reported and which they showed bind the SARS-CoV-2 spike. The definition of clonotype was according to the frequently used definition that the antibodies should use the same immunoglobulin heavy variable (IGHV) and joining (IGHJ) genes and contain heavy chain complementarity-determining region 3 (HCDR3) amino acid sequences with a minimum sequence identity of 80%. The authors should assign their antibody sequences to the specific alleles used by the antibodies to more precisely be able to define clonal relationships and to more precisely define SHM (shown in Figure 3). The authors focus primarily on a clonotype labeled 27-60, an IGHV3-53-using clonotype that they find in several of the donors. The fact that IGHV3-53-using antibodies are present in many donors has been shown in numerous previous papers. This class of antibody is generally described as public, and it is induced by both SARS-CoV-2 infection and by vaccination. The fact that SHM broadens the neutralizing capacity of this class of antibodies has been shown previously, including through an amino acid change at position 58, the Y58F change.

The authors generate amplicons of variable domain heavy chain (VH) and part of the first constant domain of the heavy chain (CH1) domain (VH-CH1 amplicons) from a set of vaccinated donors and they sequence the libraries using the Illumina NovaSeq platform. The Methods section lacks a description of what kit was used and this should be added. 2 x 150 bp is too short for obtaining good coverage of recombined heavy chain VDJ sequences, especially if the amplicons also contain part of the constant (CH1) domain as the authors state. 2 x 250 bp is better but still suboptimal as most merged reads will be too short to cover them full-length VH-CH1 amplicons.

In the section called NGS data processing it is stated that "The nonfunctional consensus reads were defined and filtered using the following criteria: (i) sequence length shorter than 250 base pairs...". It is unclear what analysis could be done with 250 bp reads where the amplicon is expected to be at least 500 bp. To clarify this, the authors should present the list of primers used (not just refer to a previous publication) and an illustration where these align to give amplicons that are not too long for the sequencing platform they used. Do they capture the complete 5' end V gene region? This is important for assigning the sequences to the correct V allele.

For the construction of the scFv phage display libraries, it appears that the VH and VK/VL amplicons were generated from cDNA produced from bulk RNA. This means that the heavy and light chain pairing is not natural as this can only be achieved if heavy and light chains are recovered from the same B cell. While this may be a standard way to make scFv libraries it is a rather outdated method to study antibody repertoires since high throughput single cell sequencing methods are now available. The scFv library approach therefore precludes the authors from defining antibody clonotypes in a stringent manner. It is quite possible that the antibodies with the same heavy chain V gene usage and same HCDR3 length come from multiple different clonal lineages.

Finally, the authors state that “Our findings suggest that SHM accumulation in the BCR space to broaden its specificity for unseen antigens is a counterprotective mechanism against virus variant immune escape” is likely correct but not new since the same conclusions have been drawn in other publications on SARS-CoV-2 spike-specific antibody evolution. If the authors could show how the SHM help the antibodies to acquire breadth that would provide useful information, but that would require a high resolution structure of the spike with the different antibodies, and/or the engineering and expression of specific antibody lineage variants that contain SHM in different positions, and combinations thereof, coupled with analysis of neutralizing activity against different virus variants to pin down how SHM generates breadth.

Point-by-point responses to Reviewers comments (NCOMMS-23-47421-T)

Reviewer #1

Comment 1-1] The manuscript by Park and colleagues aims to associate SHM accumulation with a mechanism against virus variant immune escape. It is potentially a good idea and good dataset but it needs substantial redesign and improvement on data visualization to support the claims.

Response 1-2] We thank you for the positive feedback about our research idea and dataset. In response to the feedback regarding data visualization, we have attached a redesigned Fig.1 to facilitate a better understanding of the main text. This new design aims to provide an overview of the comprehensive progress and results of our research. If the reviewer could specify more about the redesign and data visualization improvements needed, we are eager to follow advice.

Fig.1 after revision

Comment 1-2] Furthermore, neutralization assays with scFv might not inform the effector function of antibodies, thus, cannot totally inform immune escape as claimed by the authors.

Response 1-2] To address the reviewer's claim that antibody fragments without effector functions may not be informative on their ability to neutralize immune-escaped viral variants, we would like to explain the current scientific limitations as follows. The immune escape phenomenon is believed to occur mostly due to the accumulation of mutations on the spike protein of SARS-CoV-2¹. As the antibodies induced by infection or vaccination of ancestral strain became less or non-reactive to the mutated spike protein, the host loses the ability to neutralize viral variants. In this context, immune escape of SARS-CoV-2 is dependent on the reactivity of antibodies to the spike protein of SARS-CoV-2 variants and their ability to inhibit the interaction between the spike protein and the ACE-2 receptor on human cells.

In addition, many assays that measure Fc-dependent antibody responses are complex and have limitations in terms of their scalability and reproducibility, owing to inherent variability between donors². To elucidate the ability of the antibodies using *in vivo* models, knock-in mice with human Fc receptor genes vulnerable to SARS-CoV-2 would be an ideal model. Unfortunately, there is no infection model to date with such genetic modification². Due to this limitation, pseudo virus neutralization tests, viral neutralization assays using the Vero E6 cell line^{3,4}, and hACE2 transgenic mice and non-human primates models^{5,6} are commonly used in SARS-CoV-2 research. We performed a viral neutralization assay using the Vero E6 cell line to test the ability of expressed antibodies to neutralize SARS-CoV-2 spike protein variants. We believe this is generally accepted as a valid neutralization assay method by the scientific community.

Reviewer #2

Comment 2-1] The manuscript by Park et al. entitled "An ancestral SARS-CoV-2 vaccine induces anti-omicron variants antibodies by hypermutation" provide a clear and coherent analysis of the B-cell receptor (BCR) repertoires in patients receiving multiple doses of the original vaccine BNT162b2 against the SARS-CoV-2. The authors convincingly show that after a third boost of the BNT162b2 vaccine a significant part of the subjects also produce neutralizing antibodies against the omicron variant. Based on the data presented the authors suggest that the neutralizing antibodies against the omicron variant is most like due to hypermutations and less like due to VDJ recombination.

The study is well performed and provide significant novelty to our present day understanding

of how protective immunity arises. A number of minor details could be improved as indicated below, but otherwise I would recommend that the manuscript is accepted after minor revisions.

Response 2-1] We would like to thank the reviewer for their time and positive feedback.

Comment 2-2] In line 56-57 the authors write that the inefficiency of the vaccines has led to reduced efficacy of most commercially available monoclonal antibodies and antibodies under development against BA.1. In connection with the description with reduced efficiency of the BNT162b2 vaccine, I do not understand why the authors suddenly comment on the therapeutic antibodies under development.

Response 2-2] We followed the reviewer's advice and removed the corresponding sentence.

Since the emergence of severe acute respiratory syndrome coronavirus 2 (SARS-CoV-2), over 14 million sequences of variants have been collected and shared via the Global Initiative on Sharing All Influenza Data (GISAID)¹. While most mutations have little effect or are detrimental to the virus, a small subset of mutations may provide a selective advantage leading to a higher reproductive rate². The spike protein, a viral coat protein that mediates viral attachment to host cells and fusion between the virus and the cell membrane, is the primary target of neutralizing antibodies³. Serological analysis has shown that the receptor-binding domain (RBD) of the spike protein is the target of 90% of neutralizing activity in the immune sera^{4,5}. In this context, the RBD has become the essential component of most mRNA-, adenovirus-, and recombinant protein-based vaccines⁶. However, Omicron variant BA.1 has accumulated 15 mutations in RBD⁷, resulting in a 22-fold reduction in neutralization by plasma from vaccinees receiving two doses of the BNT162b2 vaccine⁸. ~~This has led to reduced efficacy of most commercially available monoclonal antibodies and antibodies under development against BA.1^{7,9}.~~ Although bivalent vaccines have been developed to overcome the immune evasion of Omicron variant⁸⁻¹¹, the majority of the population has received only monovalent vaccines to date. Fortunately, it has been proven that a third dose of the BNT162b2 monovalent vaccine can neutralize BA.1 and several recent studies have demonstrated a general increase in somatic hypermutation (SHM) of virus-specific antibodies after the third dose¹²⁻¹⁸. However, the mechanism by which Omicron variant-neutralizing antibodies are generated through repeated exposure to the ancestral spike protein remains unclear. In this study, we analyzed the chronological B-cell receptor (BCR) repertoires of BNT162b2 vaccinees and traced the development of Omicron variant-neutralizing antibodies.

Comment 2-3] The authors test all the subjects being vaccinated for whether they have had previous exposure to SARS-CoV-2 by testing whether antibodies recognizing the N-protein from SARS-CoV-2. While this is a valid approach, the data presented in extended data figure 1 lack a control. Sera from patients testing positive for SARS-CoV-2 should have been included.

Response 2-3] Following the reviewer's advice, we have added the ELISA results of a COVID-19 patient's plasma, which we used as a positive control (PC). Additionally, we have moved the data of plasma ELISA against ancestral and Omicron RBD from Fig. 1 to Extended Data Fig.2 following the reviewer #3's advice.

Extended Data Fig.1 (left; before revision, right; after revision)

Comment 2-4] In line 124 the author writes that five BCR HC clonotypes, including the 27-60, were not found in vaccinees in who the antibody clones were found. This is suggested to be due to technical issues with the Next generation sequencing. It would be good if the authors comments on how such issues may affect the general conclusions from their study.

Response 2-4] Following the comment, we added a sentence that despite the limitations of throughput of next generation sequencing, we believe that the interpretations and the conclusion would be minimally affected.

BA.1 RBD-reactive clonotypes in BCR HC repertoires

Total 246 BCR HC libraries were used to construct BCR HC clonotypes (Extended Data Fig. 5). BCR HC clonotypes of BA.1 RBD-reactive clones (BA.1 RBD-reactive BCR HC clonotype) were mapped to 293 sequences in the repertoire of 23 vaccinees (Table 1 and Supplementary Table 3). The most

frequently identified BCR HC clonotype was 27-60, found in 19 vaccinees (46%), followed clonotypes by 43-09 and 35-46, found in five and two vaccinees (12% and 5%), respectively (Supplementary Table 4). Six BCR HC clonotypes were found only in one vaccinee. Five BCR HC clonotypes, including 27-60, were not found in vaccinees in whom the antibody clones were found. This type of discrepancy is expected to inevitably originate from the incomplete *in silico* reconstitution of the BCR HC repertoire due to the limitation of the throughput of next-generation sequencing²⁸. In this regard, the BCR HC clonotypes are expected to be present in a larger population than we observed herein. **However, we also believe that this limitation would minimally affect the interpretations and conclusions drawn from our results.**

Comment 2-5] In line 168 and forward, the authors describe the construction of a scFv phage display library, where they arbitrarily selected 672 phage clones and tested their reactivity to both RBDs. The library contained randomized residues and a residue varied between A and V, thus the library should contain around 800 different phage antibody clones. To what extent does the 672 selected phage clones represent different antibodies?

In line 169 the author describes that 30 % of the selected phage antibodies bind the ancestral RBD but only 18 the BA.1 RBD – could the authors comment on this taking into account my comment made above.

Response 2-5] Following the reviewer's advice, we performed an additional phage ELISA to reach 856 randomly selected clones exceeding the hypothetical complexity of 800 (20 X 20 X 2). We have modified Fig. 2c to include this additional data. The positive rate to ancestral RBD and BA.1 RBD was changed from 30% and 3% to 31% and 2%, respectively. We believe although this data is valuable, the conclusion remains consistent with our previous findings.

To analyze the influence of the HCDR3 sequence on reactivity to ancestral and BA.1 RBDs, we generated a scFv phage display with the randomized HCDR3 sequence of ARDLXX(A/V)GGMDV, arbitrarily selected 856 phage clones and checked their reactivity to both RBDs. For the ancestral RBD, 267 phage clones (31%) showed positive reactivity (Fig. 2c, Extended Data Fig. 7 and Supplementary Table 5). However, for the BA.1 RBD, only 18 phage clones (2%) were reactive.

Fig. 2c after revision

Reviewer #3

Comment 3-1] This study addresses an interesting question, namely why a third dose of the BNT162b2 covid-19 mRNA vaccine, which encodes the ancestral SARS-CoV-2 spike protein, stimulates the production of antibodies that are capable of neutralizing Omicron subvariants. This is a well-known observation and the authors cite references 14, 15, 16, 17 and 18 to this effect. They state that the mechanism by which Omicron-neutralizing antibodies are generated following repeated vaccinations with the ancestral spike remains unclear. However, this is not quite true as there are previous publications on this topic, which the authors should include in their reference list (along with qualifying their statement) such as these:

Rapid Hypermutation B Cell Trajectory Recruits Previously Primed B Cells Upon Third SARS-Cov-2 mRNA Vaccination - PubMed ([nih.gov](https://pubmed.ncbi.nlm.nih.gov/))

Cross neutralization of SARS-CoV-2 omicron subvariants after repeated doses of COVID-19 mRNA vaccines - PubMed ([nih.gov](https://pubmed.ncbi.nlm.nih.gov/))

Response 3-1] We appreciate the reviewer's suggestion and added the recommended references accordingly. But we also would like to emphasize that our findings are still novel.

In brief, the second article only demonstrated that antibody titers were elevated after the third dose of vaccination.⁷ The first article demonstrated the numerical increase in somatic hypermutation of BCR repertoires after the third dose⁸. From the existing literatures, it remained unclear that the viral variant-binding antibody clones evolve through the accumulation of somatic mutations on antibody clones specifically reactive to the wild type virus. In our study, we demonstrate that daughter clones from a single mother BCR clone non-reactive to viral variants can achieve reactivity through the accumulation of somatic hypermutations. This novel finding was proven at the nucleotide sequence level, following methods similar to the references already cited in our manuscript.

Comment 3-2] The authors collect blood from different time points following vaccination and show plasma ELISA results in Figure 1. This type of data has been shown in several previous papers and should be moved to supplementary materials.

Response 3-2] Following the reviewer's advice, we have moved this data to supplementary materials. In the results, we redesigned Fig. 1 to a comprehensive schematic overview of the study following Reviewer 1's advice.

Before revision

After revision

Comment 3-3] They then selected six vaccine recipients that were vaccinated three times with the BNT162b2 mRNA vaccine and generated single-chain variable fragment (scFv) phage display libraries that were sequenced by Next Generation Sequencing on the Illumina NovaSeq platform. They use CoV-Ab Dab and panning to identify spike-specific antibody clonotypes in the B cell receptor repertoire data. They focus on 9 clonotypes, all of which were similar to Ab clonotypes that were previously reported and which they showed bind the SARS-CoV-2 spike. The definition of clonotype was according to the frequently used definition that the antibodies should use the same immunoglobulin heavy variable (IGHV) and joining (IGHJ) genes and contain heavy chain complementarity-determining region 3 (HCDR3) amino acid sequences with a minimum sequence identity of 80%. The authors should assign their antibody sequences to the specific alleles used by the antibodies to more precisely be able to define clonal relationships and to more precisely define SHM (shown in Figure 3).

Response 3-3] Following the reviewer's advice, we have added the data about the alleles to Extended Data Tables 2, 3, 6 and 9. The addition of this data did not change the results of our analyses and conclusions.

Comment 3-4] The authors focus primarily on a clonotype labeled 27-60, an IGHV3-53-using clonotype that they find in several of the donors. The fact that IGHV3-53-using antibodies are present in many donors has been shown in numerous previous papers. This class of antibody is generally described as public, and it is induced by both SARS-CoV-2 infection and by vaccination. The fact that SHM broadens the neutralizing capacity of this class of antibodies has been shown previously, including through an amino acid change at position 58, the Y58F change.

Response 3-4] Antibodies using IGHV3-53/3-66 have been frequently found in COVID-19 patients and vaccinees, but these antibodies are reported to have limited somatic hypermutations (SHMs)^{9,10}. Additionally, there are no reports suggesting that these antibodies exhibit broadened binding activity against variants through SHM. The Y58F mutation is also reported as a commonly occurring SHM in antibodies using IGHV3-53/3-66¹¹⁻¹³. However, currently there is no research publication comparing the reactivity for variants before and after obtaining this mutation.

Comment 3-5] The authors generate amplicons of variable domain heavy chain (VH) and part of the first constant domain of the heavy chain (CH1) domain (VH-CH1 amplicons) from

a set of vaccinated donors and they sequence the libraries using the Illumina NovaSeq platform. The Methods section lacks a description of what kit was used and this should be added. 2 x 150 bp is too short for obtaining good coverage of recombined heavy chain VDJ sequences, especially if the amplicons also contain part of the constant (CH1) domain as the authors state. 2 x 250 bp is better but still suboptimal as most merged reads will be too short to cover them full-length VH-CH1 amplicons.

Response 3-5] Following the advice, we revised the methods section. Also, 2 x 250 bp was actually sufficient for the purposes of our study, as we only sequenced the 5' region of the CH1 domain, which was sufficient to determine the subtype of BCR as described in our previous report¹⁴.

Genes encoding the variable domain of the heavy chain (V_H) and part of the first constant domain of the heavy chain (CH1) domain (V_H -CH1) were amplified using specific primers, as described previously²³. All primers used are listed in Supplementary Table 10. Briefly, total RNA was used as a template to synthesize complementary DNA (cDNA) using the SuperScript IV First-Strand Synthesis System (Invitrogen) with specific primers (primer No. 1–8) targeting the CH1 domain of each isotype (IgM, IgD, IgG, IgA, and IgE) according to the manufacturer's protocol. After cDNA synthesis, 1.8 volumes of SPRI beads (Beckman Coulter, AMPure XP) were used to purify cDNA, which was eluted in 35 μ l of water. The purified cDNA (15 μ l) was subjected to second-strand synthesis in a 25 μ l reaction volume using IGHV gene-specific primers (primer No.9–14) and a KAPA Biosystems kit (Roche, KAPA HiFi HotStart). The PCR conditions were as follows: 95°C for 3 min, 98°C for 30 s, 60°C for 45 s, and 72°C for 6 min. After second-strand synthesis, dsDNA was purified using 1 volume of SPRI beads, as described above. V_H -CH1 genes were amplified using purified dsDNA (15 μ l) in a 25 μ l reaction volume using primers containing indexing sequences (primers 15 and 16) and a KAPA Biosystems kit. The PCR conditions were as follows: 95°C for 3 min; 25 cycles of 98°C for 30 s, 60°C for 30 s, and 72°C for 1 min; and 72°C for 5 min. PCR products were subjected to electrophoresis on a 1.5% agarose gel and purified using a QIAquick gel extraction kit (QIAGEN Inc.) according to the manufacturer's instructions. The gel-purified PCR products were purified again using 1 volume of SPRI beads and eluted in 20 μ l water. The SPRI-purified sequencing libraries were quantified with a 4200 TapeStation System (Agilent Technologies) using a D1000 ScreenTape assay and subjected to next-generation sequencing on the **Illumina NovaSeq 6000 250PE (SP Chip) platform using Novaseq 6000 reagent kits (Illumina Inc.)**.

Comment 3-6] In the section called NGS data processing it is stated that "The nonfunctional consensus reads were defined and filtered using the following criteria: (i) sequence length shorter than 250 base pairs...". It is unclear what analysis could be done with 250 bp reads

where the amplicon is expected to be at least 500 bp. To clarify this, the authors should present the list of primers used (not just refer to a previous publication) and an illustration where these align to give amplicons that are not too long for the sequencing platform they used. Do they capture the complete 5' end V gene region? This is important for assigning the sequences to the correct V allele.

Response 3-6] This section of the reviewer's comments included three points for revision.

The first advice was about the length of the amplicon, which we believe is now adequately addressed in our response to the prior comment.

Following the second advice, we added an illustration to Extended Data Fig.5 and the primer binding starting sites of the list of primers in Extended Data Table 10.

Extended Data Fig.5 after revision

We also appreciate the third advice. However, we found that even though our amplicons did not cover the very 5' end of the VH genes, the rest of the sequence was satisfactory to allocate the BCR sequences into corresponding alleles.

Comment 3-7] For the construction of the scFv phage display libraries, it appears that the VH and VK/VL amplicons were generated from cDNA produced from bulk RNA. This means that the heavy and light chain pairing is not natural as this can only be achieved if heavy and light chains are recovered from the same B cell. While this may be a standard way to make scFv libraries it is a rather outdated method to study antibody repertoires since high throughput single cell sequencing methods are now available. The scFv library approach therefore precludes the authors from defining antibody clonotypes in a stringent manner. It is quite possible that the antibodies with the same heavy chain V gene usage and same HCDR3 length come from multiple different clonal lineages.

Response 3-7] We agree on the reviewer's comment about the limitations of our study. However, we would like to emphasize that the throughput of BCR sequences obtained from 246 blood samples from 41 vaccinees exceed at least 5 ml. We believe that this level of throughput was essential to obtain our findings that the daughter clones from a single mother BCR clone non-reactive to viral variants achieve reactivity through the accumulation of somatic hypermutations at the nucleotide sequence level with a lineage tree. Furthermore, as per our response to the comment from reviewer 2 (Response 2-4), we believe that a higher throughput would be helpful to draw a more detailed phylogenetic tree. The current throughput of single cell sequencing typically provides tens of thousands of BCR sequences per sample^{15,16}. The generation of heavy chain and light chain matched data from single cell sequencing at the throughput of our study would warrant an enormous budget, and we believe the obtained information would fail to rationalize the additional cost. In this regard, many prior studies have defined BCR clonotypes based on heavy chain sequences¹⁷⁻²⁰ and drew phylogenetic trees based on these findings²¹⁻²⁵.

Comment 3-8] Finally, the authors state that "Our findings suggest that SHM accumulation in the BCR space to broaden its specificity for unseen antigens is a counterprotective mechanism against virus variant immune escape" is likely correct but not new since the same conclusions have been drawn in other publications on SARS-CoV-2 spike-specific antibody evolution. If the authors could show how the SHM help the antibodies to acquire breadth that would provide useful information, but that would require a high resolution structure of the spike with the different antibodies, and/or the engineering and expression of

specific antibody lineage variants that contain SHM in different positions, and combinations thereof, coupled with analysis of neutralizing activity against different virus variants to pin down how SHM generates breadth.

Response 3-8] We appreciate the reviewer's kind suggestions for our further research. We think that accomplishing this task would take extensive time and resources, and believe that our manuscript already presents a sufficient number of novel findings that would be valuable to share with other colleagues before accomplishing the recommended tasks.

References

- 1 Harvey, W. T. *et al.* SARS-CoV-2 variants, spike mutations and immune escape. *Nature Reviews Microbiology* **19**, 409-424 (2021). <https://doi.org/10.1038/s41579-021-00573-0>
- 2 Zhang, A. *et al.* Beyond neutralization: Fc-dependent antibody effector functions in SARS-CoV-2 infection. *Nature Reviews Immunology* **23**, 381-396 (2023). <https://doi.org/10.1038/s41577-022-00813-1>
- 3 Nie, J. *et al.* Quantification of SARS-CoV-2 neutralizing antibody by a pseudotyped virus-based assay. *Nature Protocols* **15**, 3699-3715 (2020). <https://doi.org/10.1038/s41596-020-0394-5>
- 4 Bewley, K. R. *et al.* Quantification of SARS-CoV-2 neutralizing antibody by wild-type plaque reduction neutralization, microneutralization and pseudotyped virus neutralization assays. *Nature Protocols* **16**, 3114-3140 (2021). <https://doi.org/10.1038/s41596-021-00536-y>
- 5 Rosa, R. B. *et al.* In Vitro and In Vivo Models for Studying SARS-CoV-2, the Etiological Agent Responsible for COVID-19 Pandemic. *Viruses* **13**, 379 (2021). <https://doi.org/10.3390/v13030379>
- 6 Chu, H., Chan, J. F.-W. & Yuen, K.-Y. Animal models in SARS-CoV-2 research. *Nature Methods* **19**, 392-394 (2022). <https://doi.org/10.1038/s41592-022-01447-w>
- 7 Luczkowiak, J. *et al.* Cross neutralization of SARS-CoV-2 omicron subvariants after repeated doses of COVID-19 mRNA vaccines. *Journal of Medical Virology* **95** (2023). <https://doi.org/10.1002/jmv.28268>
- 8 Paschold, L. *et al.* Rapid Hypermutation B Cell Trajectory Recruits Previously Primed B Cells Upon Third SARS-Cov-2 mRNA Vaccination. *Front Immunol* **13**, 876306 (2022). <https://doi.org/10.3389/fimmu.2022.876306>
- 9 Yuan, M. *et al.* Structural basis of a shared antibody response to SARS-CoV-2. *Science* **369**, 1119-1123 (2020). <https://doi.org/10.1126/science.abd2321>
- 10 Yan, Q. *et al.* Germline IGHV3-53-encoded RBD-targeting neutralizing antibodies are commonly present in the antibody repertoires of COVID-19 patients. *Emerging Microbes & Infections* **10**, 1097-1111 (2021). <https://doi.org/10.1080/22221751.2021.1925594>
- 11 Tan, T. J. C. *et al.* Sequence signatures of two public antibody clonotypes that bind SARS-CoV-2 receptor binding domain. *Nature Communications* **12** (2021). <https://doi.org/10.1038/s41467-021-24123-7>
- 12 Chiang, H.-L. *et al.* Broadly neutralizing human antibodies against Omicron subvariants of SARS-CoV-2. *Journal of Biomedical Science* **30** (2023). <https://doi.org/10.1186/s12929-023-00955-x>
- 13 Yu, H. S. *et al.* Somatic hypermutated antibodies isolated from SARS-CoV-2 Delta infected patients cross-neutralize heterologous variants. *Nature Communications* **14** (2023). <https://doi.org/ARTN 105810.1038/s41467-023-36761-0>
- 14 Kim, S. I. *et al.* Stereotypic neutralizing V-H antibodies against SARS-CoV-2 spike protein receptor binding domain in patients with COVID-19 and healthy individuals. *Sci Transl Med* **13** (2021). <https://doi.org/ARTN eabd699010.1126/scitranslmed.abd6990>
- 15 Goldstein, L. D. *et al.* Massively parallel single-cell B-cell receptor sequencing enables rapid discovery of diverse antigen-reactive antibodies. *Communications Biology* **2** (2019). <https://doi.org/10.1038/s42003-019-0551-y>
- 16 Shapiro, E., Biezuner, T. & Linnarsson, S. Single-cell sequencing-based technologies will revolutionize whole-organism science. *Nature Reviews Genetics* **14**, 618-630 (2013). <https://doi.org/10.1038/nrg3542>
- 17 Soto, C. *et al.* High frequency of shared clonotypes in human B cell receptor repertoires. *Nature* **566**, 398-402 (2019). <https://doi.org/10.1038/s41586-019-0934-8>
- 18 Schultheiß, C. *et al.* Next-Generation Sequencing of T and B Cell Receptor Repertoires from COVID-19 Patients Showed Signatures Associated with Severity of Disease. *Immunity* **53**, 442-455.e444 (2020). <https://doi.org/10.1016/j.immuni.2020.06.024>
- 19 Turner, J. S. *et al.* SARS-CoV-2 mRNA vaccines induce persistent human germinal centre responses. *Nature* **596**, 109-113 (2021).
- 20 Sofou, E. *et al.* Clonotype definitions for immunogenetic studies: proposals from the EuroClonality NGS Working Group. *Leukemia* **37**, 1750-1752 (2023). <https://doi.org/10.1038/s41375-023-01952-7>
- 21 Turner, K. B. *et al.* Next-Generation Sequencing of a Single Domain Antibody Repertoire

- Reveals Quality of Phage Display Selected Candidates. *Plos One* **11** (2016).
[https://doi.org/ARTN e014939310.1371/journal.pone.0149393](https://doi.org/ARTN%20e014939310.1371/journal.pone.0149393)
- 22 de Bourcy, C. F. *et al.* Phylogenetic analysis of the human antibody repertoire reveals quantitative signatures of immune senescence and aging. *Proc Natl Acad Sci U S A* **114**, 1105-1110 (2017). <https://doi.org/10.1073/pnas.1617959114>
- 23 Hoehn, K. B., Lunter, G. & Pybus, O. G. A Phylogenetic Codon Substitution Model for Antibody Lineages. *Genetics* **206**, 417-427 (2017).
<https://doi.org/10.1534/genetics.116.196303>
- 24 Hoehn, K. B. *et al.* Repertoire-wide phylogenetic models of B cell molecular evolution reveal evolutionary signatures of aging and vaccination. *Proceedings of the National Academy of Sciences* **116**, 22664-22672 (2019). <https://doi.org/10.1073/pnas.1906020116>
- 25 Chernyshev, M. *et al.* Vaccination of SARS-CoV-2-infected individuals expands a broad range of clonally diverse affinity-matured B cell lineages. *Nat Commun* **14**, 2249 (2023).
<https://doi.org/10.1038/s41467-023-37972-1>

Reviewers' Comments:

Reviewer #2:

Remarks to the Author:

The authors have revised the manuscript in a satisfactory manner, addressing the initial concerns I had.

So based on the revised manuscript I can recommend acceptance of the manuscript for publication

Reviewer #3:

Remarks to the Author:

While the authors have added information that improves the transparency of the methods used, my main concerns remain, ie that the technology used is not optimal for studies of human antibody repertoires and antibody lineage evolution.

I had asked the authors to assign the antibody sequences to the specific alleles used by the antibodies for more precise definitions of clonal relationships and SHM. I also asked them to present a list of primers used and an illustration showing where the primers align. Finally, I asked if the primers allow the complete 5' end V gene regions to be captured, which is important for assigning the sequences to the correct V alleles.

The authors have now added a list of primers and an illustration of where the primers align (New Extended Data Fig.5). From this illustration it is clear that the primers align to regions within the coding sequence of the V genes, which means that the full V gene regions are not captured. This makes allele assignments inaccurate as some polymorphisms may be present outside of the regions contained within the amplicons.

Therefore, at the very least, the authors should add a "limitations of the study" section to the paper where it is clearly stated that the allele assignments are not precise since the full V genes were not sequenced. Furthermore, very few primers were used per gene family, which means that some alleles may be missed. These limitations should be stated clearly.

Point-by-point responses to Reviewers comments (NCOMMS-23-47421A-Z)

Reviewer #2

Comment 2-1] The authors have revised the manuscript in a satisfactory manner, addressing the initial concerns I had.

So based on the revised manuscript I can recommend acceptance of the manuscript for publication

Response 2-1] We would like to thank the reviewer's satisfaction with our response and for supporting the decision to publish our research.

Reviewer #3

Comment 3-1] While the authors have added information that improves the transparency of the methods used, my main concerns remain, ie that the technology used is not optimal for studies of human antibody repertoires and antibody lineage evolution.

I had asked the authors to assign the antibody sequences to the specific alleles used by the antibodies for more precise definitions of clonal relationships and SHM. I also asked them to present a list of primers used and an illustration showing where the primers align. Finally, I asked if the primers allow the complete 5' end V gene regions to be captured, which is important for assigning the sequences to the correct V alleles.

The authors have now added a list of primers and an illustration of where the primers align (New Extended Data Fig.5). From this illustration it is clear that the primers align to regions within the coding sequence of the V genes, which means that the full V gene regions are not captured. This makes allele assignments inaccurate as some polymorphisms may be present outside of the regions contained within the amplicons.

Therefore, at the very least, the authors should add a "limitations of the study" section to the paper where it is clearly stated that the allele assignments are not precise since the full V genes were not sequenced.

Furthermore, very few primers were used per gene family, which means that some alleles may be missed. These limitations should be stated clearly.

Response 3-1] We appreciate the reviewer's comments. Below is our response addressing the concerns raised by the reviewer:

1. Concerns about primer coverage of IGHV alleles

The primer set utilized in our study was referenced from the European BIOMED-2 collaborative study, based on its primer design and standardization of high quality and reproducible immunoglobulin analyses¹. This primer set and protocol, having amassed 3588 citations to date, are also being actively utilized by other research groups investigating BCR repertoire profiling, such as Scott D. Boyd's group² and Dennis R. Burton's group³.

To assess whether our set of six forward primers can anneal and amplify IGHV alleles, we used the most recently updated (02 October, 2023) IGHV allele sequences from the IMGT database (<https://www.imgt.org/>). We excluded the ORF, partial sequences, and pseudogenes, resulting in a total of 270 alleles for analysis. Among these alleles, we found that with three mismatches, 269 alleles (99.63%) could be amplified, and with four mismatches, all alleles could be included (Supplementary Data 7). Among a total of 270 alleles, we confirmed that our primers can facilitate successful amplification in 269 alleles, as there are no mismatches with the primer sequences up to the last 6 nucleotides at the 3' end⁴. For the IGHV4-39*08 allele, although there is only one mismatch with the primer, its position at the third nucleotide from the 3' end makes amplification unlikely (Supplementary Data 7).

2. Concerns regarding inaccurate IGHV allele annotation

Following our confirmation of the effectiveness of our primers in amplifying whole alleles, we considered the concerns raised by the reviewer regarding potential inaccuracies in allele annotation (excel 1). Upon closer examination, it was found that the likelihood of inaccurate allele annotation was approximately 12.96% (35 out of 270), which we believe is not high enough to cause major concerns. This low likelihood may be attributed to the fact that the primers used in our study bind relatively closely to the IGHV start site (18-47nt), allowing for a high probability of accurate allele annotations, due to the extended lengths of readable IGHV sequences.

Subsequently, we confirmed the identification of 187 out of 270 alleles (69.3%) across 246 BCR HC libraries (Supplementary Data 9). Considering that all vaccinees are of Korean

descent, we sought to investigate the potential presence of restricted allelic and copy number variations based on race. Specifically, we examined the distribution and prevalence of alleles unique to East Asians, anticipating variations influenced by ethnicity⁵. Analyzing cohorts from three Chinese populations (CDX: Chinese Dai in Xishuangbanna, China; CHB: Han Chinese in Beijing, China; CHS: Han Chinese South), one Japanese population (JPT: Japanese in Tokyo, Japan), and one Vietnamese population (KHV: Kinh in Ho Chi Minh City, Vietnam), we observed an average of 62.7% of detected alleles (table below), a figure closely resembling our findings.

Cohort	CDX	CHB	CHS	JPT	KHV
The number of samples	3	7	23	2	47
The number of total alleles	194	194	194	194	194
The number of alleles detected	121	119	121	120	127
percentage of detected alleles (%)	62.37	61.34	62.37	61.86	65.46

It seems the reviewer was concerned about the potential for errors when calculating the number of SHMs in relation to the main findings of the paper, particularly in cases where direct comparison with the germline allele was not conducted. We were mindful of this aspect and took it into consideration. We mitigated this concern by counting the number of SHMs only in the sequenced regions amplified by our primers, compared to the germline sequence. It's worth noting that this approach ensures greater accuracy, as the SHM counted regions are identical in sequence, regardless of differences in actual allele germlines.

Thus, we believe that the primer set utilized in our study did not pose significant issues in conducting the study and asserting our findings. To ensure accurate comprehension by readers, we have included the following statement in the Next-generation sequencing and NGS data processing sections of the methods:

Methods

Next-generation sequencing

Genes encoding the variable domain of the heavy chain (V_H) and part of the first constant domain of the heavy chain (CH1) domain (V_H -CH1) were amplified using specific primers,

as described previously²³. All primers used are listed in Supplementary Data 6. It is expected that these six IGHV-specific primers can amplify a total of 269 out of 270 IGHV alleles (99.63%) available in the IMGT database. *IGHV4-39*08* is expected to encounter difficulty in amplification due to a mismatch with the germline sequence at the third nucleotide from the 3' end of the primer³² (Supplementary Data 7).

NGS data processing

Raw sequencing reads obtained from sequencing the V_H-CH1 region of B cells in peripheral blood were processed using a custom pipeline. The pipeline included adapter trimming and quality filtering; unique molecular identifier (UMI) processing; V(D)J gene annotation; clustering; quality control; and diversity analysis.

The forward reads (R1) and reverse reads (R2) of the raw NGS data were merged using paired-end read merger (PEAR) v0.9.10 with the default settings³³. The merged reads were q-filtered under the q20p95 condition, resulting in 95% of base pairs in the reads having a Phred score greater than 20. Primer positions were identified in the quality-filtered reads, and primer regions were trimmed to remove the effects of primer synthesis errors while allowing one substitution or deletion. We identified 187 out of 270 IGHV alleles (69.3%) from the IMGT database across six time points in 41 vaccinees (246 BCR HC libraries). Our vaccinees are of Korean nationality. This ratio indicates a resemblance to the average allele distribution observed among Asian populations (62.7%)³⁴, since the vaccinees in this study are of Korean nationality (Supplementary Data 9).

.....

Among the annotation results, the IGHV genes, IGHJ genes, HCDR3 sequences, and number of SHMs were extracted for further analysis. The number of SHMs cannot be determined for sequences outside the primer binding sites. Therefore, we compared the amplified regions

from primer-mediated amplification with the IGHV gene germline sequence to calculate SHMs. The nonfunctional consensus reads were defined and filtered using the following criteria: (i) sequence length shorter than 250 base pairs from each R1 and R2, (ii) presence of a stop codon or frameshift in the entire amino acid sequence, (iii) failure to annotate one or more HCDR1, HCDR2 and HCDR3 regions, and (iv) failure to annotate the isotype.

References

- 1 Van Dongen, J. J. M., et al. "Design and standardization of PCR primers and protocols for detection of clonal immunoglobulin and T-cell receptor gene recombinations in suspect lymphoproliferations: report of the BIOMED-2 Concerted Action BMH4-CT98-3936." *Leukemia* 17.12 (2003): 2257-2317
- 2 Nielsen, Sandra CA, et al. "Human B cell clonal expansion and convergent antibody responses to SARS-CoV-2." *Cell host & microbe* 28.4 (2020): 516-525
- 3 Briney, Bryan, et al. "Commonality despite exceptional diversity in the baseline human antibody repertoire." *Nature* 566.7744 (2019)
- 4 Stadhouders, R. et al. The Effect of Primer-Template Mismatches on the Detection and Quantification of Nucleic Acids Using the 5' Nuclease Assay. *J Mol Diagn* 12, 109-117 (2010). <https://doi.org:10.2353/jmoldx.2010.090035>
- 5 Khatri, I. et al. Population matched (pm) germline allelic variants of immunoglobulin (IG) loci: Relevance in infectious diseases and vaccination studies in human populations. *Genes & Immunity* 22, 172-186 (2021). <https://doi.org:10.1038/s41435-021-00143-7>

IGHV	allele count	min mismatch among alleles*	un-discriminated_alleles
IGHV1-18	3	1	
IGHV1-2	8	1	
IGHV1-24	1		
IGHV1-3	4	0	1 (IGHV1-3*01, IGHV1-3*05)
IGHV1-45	3	1	
IGHV1-46	4	1	
IGHV1-58	2	1	
IGHV1-69	17	0	5 (IGHV1-69*01, IGHV1-69*12 IGHV1-69*01, IGHV1-69*13 IGHV1-69*04, IGHV1-69*09 IGHV1-69*06, IGHV1-69*14 IGHV1-69*12, IGHV1-69*13)
IGHV1-69-2	1		
IGHV1-69D	2	1	
IGHV1-8	3	1	
IGHV2-26	5	0	1 (IGHV2-26*01, IGHV2-26*05)
IGHV2-5	6	0	3 (IGHV2-5*05, IGHV2-5*06 IGHV2-5*05, IGHV2-5*09 IGHV2-5*06, IGHV2-5*09)
IGHV2-70	15	0	2 (IGHV2-70*11, IGHV2-70*15 IGHV2-70*16, IGHV2-70*17)
IGHV2-70D	2	1	
IGHV3-11	5	0	1 (IGHV3-11*03, IGHV3-11*05)
IGHV3-13	6	1	
IGHV3-15	9	0	3 (IGHV3-15*01, IGHV3-15*02 IGHV3-15*01, IGHV3-15*09 IGHV3-15*02, IGHV3-15*09)
IGHV3-20	2	1	
IGHV3-21	7	0	1 (IGHV3-21*01, IGHV3-21*02)
IGHV3-23	4	0	1 (IGHV3-23*01, IGHV3-23*04)
IGHV3-23D	1		
IGHV3-30	20	1	
IGHV3-30-3	3	1	
IGHV3-30-5	3	3	
IGHV3-33	8	1	
IGHV3-35	1		
IGHV3-43	2	3	
IGHV3-43D	4	0	3 (IGHV3-43D*04, IGHV3-43D*05 IGHV3-43D*04, IGHV3-43D*05_2 IGHV3-43D*05, IGHV3-43D*05_2)
IGHV3-48	4	1	
IGHV3-49	5	0	1 (IGHV3-49*03, IGHV3-49*05)
IGHV3-53	6	0	2 (IGHV3-53*01, IGHV3-53*02 IGHV3-53*04, IGHV3-53*06)
IGHV3-62	1		
IGHV3-64	6	0	1 (IGHV3-64*02, IGHV3-64*07)
IGHV3-64D	3	1	
IGHV3-66	4	1	
IGHV3-7	5	1	
IGHV3-72	1		
IGHV3-73	2	0	1 (IGHV3-73*01, IGHV3-73*02)
IGHV3-74	3	0	1 (IGHV3-74*01, IGHV3-74*02)
IGHV3-9	4	1	
IGHV3-NL1	1		
IGHV4-28	7	0	2 (IGHV4-28*01, IGHV4-28*07 IGHV4-28*02, IGHV4-28*05)
IGHV4-30-2	5	1	
IGHV4-30-4	5	1	
IGHV4-31	5	1	
IGHV4-34	8	0	1 (IGHV4-34*01, IGHV4-34*02)
IGHV4-38-2	3	1	
IGHV4-39	6	0	1 (IGHV4-39*01, IGHV4-39*08)
IGHV4-4	6	1	
IGHV4-59	6	0	1 (IGHV4-59*01, IGHV4-59*07)
IGHV4-61	8	1	
IGHV5-10-1	4	0	1 (IGHV5-10-1*01, IGHV5-10-1*03)
IGHV5-51	5	0	1 (IGHV5-51*01, IGHV5-51*03)
IGHV6-1	2	0	1 (IGHV6-1*01, IGHV6-1*02)
IGHV7-4-1	4	1	

Not discriminated alleles	whole alleles	Cannot discriminating case (%)
35	270	12.96

* The number of mismatches were calculated based on the **V gene region that our primers could cover[†]** (see below figure)